# Impact of Covid-19 on informal employment: A case study of women domestic workers in Khyber Pakhtunkhwa, Pakistan

Adnan Ahmad Dogar[1], Ikram Shah[2]*, Tahir Mahmood[3], Noor Elahi[2], Arif Alam[2], Urooj Gul Jadoon[2]

1 Department of Tourism, Hospitality and Hotel Management, Kohsar University Muree, Punjab, Pakistan, 2 Department of Development Studies, COMSATS University Islamabad, Abbottabad Campus, Abbottabad, Pakistan, 3 Department of Sociology and Rural Development, Karakoram International University, Chilas Campus, Chilas, Gilgit Baltistan, Pakistan

These authors contributed equally to this work.
* ikramshah@cuiatd.edu.pk

**Data Availability Statement:** All relevant data are within the manuscript and its Supporting Information files.

## Abstract

COVID-19 indiscriminately impacted all the segments of the global society. Due to the unstructured job market, women in the informal sector were at high risk to experience the adverse effects of the pandemic. This paper aims to explore the impact of COVID-19 on women domestic workers and their families. Semi structured interviews conducted with fifty-four women domestic workers in three districts of Khyber Pakhtunkhwa, Pakistan were analyzed in five themes: disruption caused by lockdown, loss of livelihood, economic hardships, social support mechanism, and challenges faced during the pandemic. The study underlines that the pandemic left severe impacts on access to basic services, employment, food security at household level and pattern of expenditures. The plummeting economic activities led to sudden drops in earnings that forced families to sell their assets and incur debts. Respondents lamented over the social support system and considered it a necessary but not sufficient condition for uplifting the lives of the poor. Strains in marital relationships led to stress, anxiety and domestic violence among families. The utmost concern was the restoration of economic activities and urgent policy interventions to strengthen social safety measures for the vulnerable segments of society.

## Introduction

COVID-19 pandemic, which broke out in Wuhan city of China in late 2019 [1], has directly and indirectly affected every sphere of life across the world [2–7]. At the time of writing this article, the virus has afflicted 130 million people and caused more than two million fatalities worldwide (as of August 2020). In June 2020, WHO announced COVID-19 guidelines for wearing masks and other protective measures for the general masses to protect themselves from the adverse effects of the virus [8]. Although the World Health Organization (WHO), United Nations (UN) agencies, many multilateral and bilateral organizations, governments

**Funding:** The authors received no specific funding for the work.

**Competing interests:** The authors have declared that no competing interests exist.

institutions, academia, and research institutions are findings ways to control and minimize the effects of COVID-19, but as time passes the spread of virus leads to more challenges and panic. The impacts of COVID-19 on health sectors, economy, supply chain, labor market, and home life are manifold and might last for a long time [9–12]. After the pandemic, changes in the workplaces and jobs have been significantly observed throughout the globe. The United Nations Industrial Development Organization (UNIDO) reported that the global workforce has declined at an unprecedented rate [13]. World Travel and Tourism Council (WTTC) reported that as the arrival of international tourists dropped by 22% in the first quarter of the year and is expected to decline further by the end of the year, more than 98 to 200 million jobs are at risk [14, 15] while millions of jobs are in jeopardy worldwide in various sectors. Those who have been fortunate enough and engaged in the workforce, their nature of work has significantly changed with unprecedented remote work [16]. For example, majority of the employers advised their employees permanent work-from-home [17–19]. The pandemic has far-reaching implications for labor market participation and has reached a staggering level that threatens further unemployment in the world [20]. For instance, the US president indicated that foreign workers would struggle in finding jobs [21]. Changes in the nature of work have cascading impacts on job outcomes such as productivity, organizational commitment, and job satisfaction.

COVID-19 has indiscriminately impacted the job market all over the world, however, the devastating impacts are more pronounced in low and middle-income countries [22]. Particularly, the workers engaged in the informal sector are at high risk of facing the adverse effects of COVID-19. According to the International Labor Organization (ILO), informal workers represent 61% of the global workers [23]. Sadly, of these, 1.6 billion informal workers will bear the unpleasant consequences due to restrictions such as lockdown, restriction on inter-and intra-state mobility and social distancing, and the resulting economic downturn will severely affect the livelihood of informal workers [24]. Majority of these informal workers belong to low- and middle-income countries which include bars and restaurants, arts, leisure and entertainment, transportation, accommodation and real estate business, travel agents and tour operators, street vendors and market traders [25]. The crisis emerged from COVID-19 threatened the informal workers' means to earn and feed their families. COVID-19 disproportionately affected these informal workers in informal sectors. They rarely received government support and attention. Therefore, around the world, unemployment increases are manifold because these workers are considered at high risk due to the adverse effects of COVID-19. In addition, the global economic crisis that emerged from COVID-19 made informal workers more vulnerable to poverty, and more shutdowns of economic activities can further deteriorate their livelihood and subsistence depending on the day-to-day struggle [26]. Of these informal workers, women are the most vulnerable segment to face the negative effects of COVID-19 [27]. Throughout the world, compared to 36.6 percent of all employed men workers, approximately forty percent of all women workers are hit the hardest economically by the pandemic [28, 29]. For instance, for COVID-19 pandemic, in New Zealand, majority of the unemployed workers are women while in United States of America (USA), most women are adversely affected due to job loss, reduced wages, and the resulting economic fallout [30, 31]. According to World Bank, informal sector has been deteriorated in South Asia which has rendered millions of informal workers vulnerable and has put their lives in the throes of numerous challenges [32]. Particularly, women engaged in the domestic work in non-essential sectors, for instance, food services, hospitality, domestic work, housemaids and childcare are at the high risk of bearing the economic crisis originated from the pandemic. Majority of these women domestic workers are migrants from rural areas to urban cities for securing their livelihood and earning opportunities [27, 29, 30].

### The impacts of COVID-19 on women domestic workers in Pakistan

Globally, two billion people are engaged in the informal sector (61 percent) of which 58 percent comprises of women only [33]. The informality of workforce exists throughout the world, however, the phenomena is more pronounced in developing countries. For instance, in India, 83 percent of the workforce is informal [34], Nepal has more than 70 percent of employees in informal sector [35] while Bangladesh has 89 percent labor force associated with informal economy [36]. In Pakistan, approximately 72 percent of the informal labor force is engaged in non-agriculture employment of which 71.8 percent is women [37]. Majority of women domestic workers associated in informal sector are engaged as housemaids or childcare labors. Most of these workers have migrated to main cities from rural areas or surrounding peripheries. According to ILO [38], in Pakistan, more than 8.5 million workers are engaged in domestic work while majority of them are uneducated with no skills. Mostly, these domestic workers provide their services as housemaids, caregivers, home cleaners, and cooks. The domestic workers suffer in many dimensions such as lack of social protection, unregulated employment, lack of decent work opportunities, absence of legal protection, long working hours, and minimum monetary benefits and earnings. In addition, during COVID-19 restrictions such as lockdowns, social distancing and other measures to avoid the potential virus transmission, majority of the households performed home chores on their own. Therefore, the unprecedented situation that emerged from the pandemic amplified their vulnerability to income and job loss, poverty, food insecurity and hand-to-mouth life. In Pakistan, the women domestic workers engaged in non-agriculture informal sectors are more vulnerable to the ravaging impacts of COVID-19. As these workers received less attention, social assistance, and support from the government, they are struggling in findings ways to meet their basic needs to feed their dependent family members. In addition, on one hand, the overall plummeting of economic activities in the country and weak social protection mechanism at the government level further contributed to the exacerbating situation of these workers. On the other hand, these domestic workers were immensely struggling for alternative coping strategies to meet their basic needs to protect themselves and their dependents from being hand-to-mouth. The current study aims to assess the impacts of COVID-19 pandemic on women engaged in domestic work such as housemaids, laundry-maids, cooks, and child caregivers. The contribution of this study is twofold. Firstly, the paper contributes to the scarce literature from the developing world to explore the impacts of pandemic on the women domestic workers engaged as housemaids, laundry-maids, cooks, and child caregivers in informal sector, how these women domestic workers managed the social and economic hardships during lockdown period, and what major challenges were faced in the crisis during the pandemic. Secondly, the article informs the government institutions in formulating effective policy initiatives to tackle the post-lockdown adversity triggered by COVID-19 pandemic particularly for informal workers.

## Material and methods

### Study area

The study was conducted in the districts of Abbottabad, Haripur, and Mansehra in Khyber Pakhtunkhwa province of Pakistan. All districts are part of the Hazara division and are the most populous of the region. District Abbottabad has a population of 1.3 million covering 1976 square kilometers, district Haripur has 10.03 million inhabitants stretching over 1725 square kilometers, while district Mansihera has 15.56 million people inhabiting an area of 4125 square kilometers [39]. All districts are home to natural forests, wildlife, mountains, major tourist destinations, industrial zones, and considered as the gateway to the Northern

areas and Azad and Jammu Kashmir. In the recent past, the population of these districts has increased rapidly due to rural-urban migration, conflicts in Federally Administered Tribal Areas (erstwhile FATA), and the frequent disasters in the neighboring districts. The district is also among those with high literacy rates in the country. As these districts are the economic hub of Hazara division, they are mostly overcrowded since the maximum possible people from the peripheries have moved to the main cities either on daily basis or along with families to search for jobs and livelihood. The inhabitants of the main cities mostly belong to affluent, educated, and working-class families, therefore, in majority cases households prefer to hire someone for household chores. Thus, the informal workers are mostly seen in the streets roaming to earn opportunities.

## Participants

The participants of this study were women workers engaged in the informal sector who are providing their services as housemaids, laundry-maids, cooks, and child caregivers at household level on daily basis for fixed hours (*see Table 1*). Many women domestic workers serve two to five households at the same time in different capacities.

## Research approach

The outcomes of the current study are based on qualitative data collection. The qualitative data was collected through semi-structured interviews, field notes and personal observations. The interview guide was translated into local language from English to capture the impact of lockdown on domestic women workers. The data collection tool was pretested, and changes were incorporated for clarity and understanding. The University's Departmental Ethical Review Committee (DERC) has approved all procedures used in this study, however, keeping in view the unprecedent situation emerged from COVID-19, the committee exempted the study from the ethical approval for the primary data while advising the researchers to take verbally informed consent from the respondents and follow the COVID-19 guidelines. Through purposive snowball sampling techniques, the women domestic workers in informal sector were approached who provided their services in different capacities to many households in daily fixed hours. The data was collected during October 2020 to February 2021. The researchers adopted all the protective measures to avoid personal interaction with respondents such as using hand gloves, wearing of facemasks, using hand sanitizers, and observing social distancing. In case the domestic workers had any ailment or symptoms, the researcher dropped them from the data collection process. The interviews were conducted in two phases. In the first phase, the locations of women domestic workers were identified for interviews and consent was sought for participation in the study. Due to the potential risk of COVID-19 infection and illiteracy among majority of domestic women workers at the beginning of the study, verbal consent was obtained from all the respondents through telephonic and physical means. The informed consent was communicated in local language with respondents to ensure that they

**Table 1. Details of respondents.**

| Category of Respondent | Percentage | Frequency |
|---|---|---|
| Housemaid | 38.88 | 21 |
| Laundry amid | 29.62 | 16 |
| Cook maid | 22.22 | 12 |
| Childcare giver | 9.28 | 5 |
| **Total** | **100** | **54** |

understood the aims of this research and to what they were consenting. In the second stage, the data collection process was completed. In total, fifty-four semi structured face to face interviews and telephonic interviews were conducted with housemaids, laundry-maids, cook maids, and child caregivers in separate sittings in their dwellings. For telephonic interviews, the contact number of respondents or other male respondents was requested. A copy of interview guide was shared with respondents for clarity and understanding of interview questions. Out of fifty-four semi structured interviews, seventeen were conducted face to face while thirty-seven were conducted telephonically. Analysis of qualitative data was carried out through thematic matrices analysis. Initially, major themes were identified in the data and sub-themes were generated. For commonality, variation, and critical aspect within the data were taken as described by Hill [2012] [40]. Descriptive statistics were conducted through Statistical Package for Social Sciences (SPSS) version 22.

## Results and discussion

### Characteristic respondents

Table 2 presents the demographic details of respondents. According to the respondents, they initially migrated to the study area from the surrounding districts to secure livelihoods and explore earning opportunities. Among respondents, thirty-seven lived in the city for the last 12

**Table 2. Demographic details of respondents.**

| Locality, Marital Status, Age, Family Size, Education | Percentage | N |
|---|---|---|
| Locality | | |
| *Abbottabad* | 37.03 | 20 |
| *Haripur* | 25.92 | 14 |
| *Mansehra* | 37.03 | 20 |
| **Total** | **100** | **54** |
| Marital Status | | |
| *Single* | 12.96 | 7 |
| *Married* | 72.22 | 39 |
| *Widowed* | 5.55 | 3 |
| *Divorced* | 11.11 | 6 |
| **Total** | **100** | **54** |
| Age | | |
| *Up to 20 Years* | 5.56 | 3 |
| *21–30 Years* | 24.07 | 13 |
| *31–40 Years* | 40.75 | 22 |
| *41–50 Years* | 20.37 | 11 |
| *51–60 Years* | 7.40 | 4 |
| *61 and above* | 1.85 | 1 |
| **Total** | **100** | **54** |
| Education | | |
| *No Schooling* | 27.78 | 15 |
| *Less than primary* | 48.14 | 26 |
| *Primary level* | 18.51 | 10 |
| *Secondary school/Vocational education* | 5.56 | 3 |
| *Higher secondary/college degree* | - | - |
| *Bachelor's degree/higher education* | - | - |
| **Total** | **100** | **54** |

years, while twenty-four respondents lived in the area for 5 to 7 years. Fifty-six respondents lived for the last two to four years while 28 respondents were relatively new in the locality. Majority of the participants (66.20%) lived in joint families while a few working women lived as a single parent family. Majority of the participants was in the age group range of 21–30 and 31–40 years old. The demographic details of respondents also highlighted their average family size is 6.8 per household and that they have either no schooling or less than primary level education.

Out of 54 respondents, 21 worked as housemaids, 16 as laundry-maids, twelve as cooks, and five as child caregivers. Number of persons who earn ranged from 1–4 person per household with each household having high level of dependent members (*see Table 3*). In majority cases, the spouses of all the respondents worked as street vendors, taxi drivers, construction labors, juice sellers and garbage collectors. In 25 households, children were also working to earn for their families. In most cases, children were enrolled in schools while only 23 respondents reported that their children were studying in government schools. Out of 54 respondents, forty-seven respondents had rented accommodation. Majority of rented houses were single room with kitchen facility while a few respondents were living in the servant quarters of the households where they worked. The rent of accommodations ranged from US $55 to $65 per month. Only five respondents had bank accounts, all the respondents had mobile phones, and 36 respondents had domestic animals for meeting their milk needs.

## Impact of COVID-19 outbreak on women domestic workers engaged in informal sector

### Disruption caused from lockdown

COVID-19 severely disrupted access to services and resources. Respondents highlighted that after the emergence of COVID-19 and subsequent lockdowns, majority of basic services were closed. For instance, the National Command and Operation Center (NCOC) directed to civil administration on the closure of economic activities, banned inter and intra city mobility, suspended educational facilities and reduced health services to a minimum. Respondents highlighted that the implementation of such kind of restrictions created a stressful environment, and everyone showed his/her concern. Particularly, those who were already suffering from limited access to services and opportunities were in trouble. Women domestic workers documented that they faced difficulty in access to basic services during lockdown.

> During lockdown, life became very challenging, because everything that you need was inaccessible and shut down. For people like us who are totally dependent on their daily earnings were in serious trouble. Mostly, we don't keep enough food items stocked at home. When the lockdown was announced, everyone rushed to the groceries stores and purchased enough food items for themselves. But people like us who don't have enough money and hardly manage their home were at risk to face severe problems to feed themselves and their dependents.

**Table 3. Descriptive statistics of demographic details.**

| Marital Status, Age, Family Size, Education | Mean | Range | S.D. |
|---|---|---|---|
| Household members | 6.8 | 5–11 | 1.41 |
| Number of children per family | 3.06 | 1–5 | 1.58 |
| Number of persons who earn | 2.42 | 1–4 | 1.17 |
| Number of family members who are dependent | 4.07 | 3–7 | 2.02 |

The disruption in social services and market closure pushed people to experience an unprecedented situation. The respondents highlighted that chaos ensued without any alternative arrangements. For instance, people were informed not to come to work due to market closure and fear of being infected by others, the suspension of basic services such as transport, education and most importantly health services generated a crisis for them which was never experienced before. One of the childcare giver women workers reported,

My mother has serious health issues; she needs frequent checkups from physicians. But during the lockdown, health services were limited to only emergency services. We visited the health facility several times but were not allowed to have an appointment. She suffered a lot and became weaker day by day. Because I could not afford the private clinic expenses, we mostly relied on old medication, but for me it was very hard to purchase medicines.

Furthermore, the closure of private clinics also made it challenging for poor families who had limited resources and access to quality health care services. These private clinics operated by healthcare practitioners provided health services relatively cheaper. Restrictions and bans on these private clinics put the poor families at disadvantage.

## Loss of livelihood

COVID-19 indiscriminately affected the livelihood of women domestic workers engaged in informal sector. The primary data revealed that in most cases, women domestic workers relied on their own earnings to support their families. Before COVID-19, these domestic workers were dependent on the households where they offered their services in different capacities for fixed hours in a day. For instance, one respondent highlighted that they mostly worked at three to four households at the same time and visited them in fixed hours. While working, the households facilitated them in kind such as food items, cloths, vegetables, fruits and supported them financially at the time of any emergency. Women domestic workers documented that the outbreak of COVID-19 and the subsequent unprecedented situation of lockdown has adversely affected their livelihood options. One of the housemaids expressed during interview:

I worked as a housemaid in three different households. Since the government announced the lockdown, I have been informed by the families not to come for work. Initially, I thought that it might be for one or two days but as the days passed, the period became longer. Hence, I was worried about our livelihood because I had no income throughout the lockdown period. In addition, at home, we don't have sufficient amount of money to meet our needs. Therefore, at that time I was struggling to find any alternative.

The interview details also revealed that COVID-19 outbreak created a sense of worry among respondents regarding their livelihood. They faced an array of problems to manage their livelihood during COVID-19. For instance, majority of the women domestic workers whose husbands earned to support their families reported that restrictions on transport services further exacerbated their livelihood opportunities.

My husband is a taxi driver and he used to go to work regularly. Since the lockdown was imposed, he could not find work. During the lockdown, he tried several times but was unable to get enough customers. During that period, he mostly returned home empty handed with zero earnings.

Domestic women workers from single parent families were adversely affected by lockdown. Most of these families were unable to pay for their daily expenses. Though they were supported by other family members such as fraternal and maternal relatives and in-laws, but they hardly met their basic needs. One respondent recalled that time and documented,

There were no food items left at home to eat. I remember one morning when I woke up early and went out to find something for breakfast. After one and half hour struggle, I knocked at a door hopelessly and requested them to give something for breakfast. Thankfully, that family gave me adequate amount of grocery items and vegetables.

Most of the respondents illustrated that they borrowed money from relatives, neighbors, and friends to meet the daily expenses. The lockdown and the subsequent crisis impoverished women domestic workers and their families. Although the intensity of these problems varies across families, but the adverse effects are more pronounced in single parent families and in families with more dependent family members.

After one week of lockdown, I was really worried how to manage the required groceries until I could resume work. Because, we have more family members, I used to buy wheat flour, sugar, edible oil etc., from a nearby grocery store at the corner of our street. But during the lockdown when shops were allowed to be open only for certain hours, I visited many times, but the owner refused to give me required food items on credit.

Majority respondents reported the loss of livelihood opportunities resulting from the shutdown of economic activities. A sense of worry was reported by all the respondents over their retention of jobs and earnings, debt repayment, and restoration of livelihoods opportunities.

### Economic hardship

The interview details revealed that majority of the respondents disproportionately experienced financial hardship during COVID-19. In most cases, women domestic workers lost their jobs and earnings which directly affected their household expenditures. Though they were working with households for many years, they received minimum financial support from the employers. The respondents reported that in many cases they were forced to leave their jobs while others lost their jobs due to unavailability of intra city transport which restricted their travel to the workplaces. The lost jobs and lack of earning opportunities resulted into debt which put the women domestic workers under pressure and vulnerable to economic hardships. For instance, majority of respondents was worried about the repayment of debt that compelled them to make different compromises. They were struggling to readjust their household expenditures and cut down on food choices, health services, leisure activities, recharging of mobile and cable television (in few cases), participation in family and social gatherings, and personal expenses. One of the women domestic workers documented that:

Having no job and minimum earnings compels you to revisit all of your life choices. We experienced it for the first time and were confused what to do when everyone around us had the similar economic conditions. The lockdown period drastically affected our economic position.

The respondents highlighted that the economic hardships vary by families and social arrangements. For instance, in the first case, those women domestic workers who had more working family members were relatively in a better position to absorb economic shocks. While

in the second case, women domestic workers who were sole bread earners or had more dependent family members were in serious economic trouble. In both cases, majority respondents agreed that economic hardships severely and adversely affected their overall wellbeing and family welfare.

> We had a cow and a goat to meet our milk needs. But I sold the goat for a very cheap price because I had no money in hand to feed my family.

The interview details also revealed that lost job and reduction in earnings affected the household dynamics such as shrinking of household earnings, withered savings, increased debt, difficulty in management of household expenditure, and changing nature of family relationships. Respondents documented that the sudden reduction in economic resources and subsequent economic constraints made the life of women domestic workers stressful and challenging.

> We hardly managed our household expenditures, and the overall quality of life has deteriorated. We faced difficulty in payment of rent, food intake got restricted, marital relationships were stressed. Due to indebtedness, families became economically under-pressure which led to financial insecurities; therefore, families were forced to sell their personal assets.

In addition, women domestic workers complained that increase in prices of basic commodities further aggravated their hardship. The pandemic and resulting lockdown created both long- and short-term economic hardships for domestic women workers. In the short term, they had to restore the household expenditures while in the long run they had to manage their economic resources in such a way that enabled them to minimize their economic insecurities and hardships.

## Financial assistance and social support during COVID-19

The government Ehsas Emergency Cash program was launched just after the introduction of lockdown. The purpose of the program was to facilitate those vulnerable groups of society who were experiencing economic hardship during the pandemic. Government allocated 203 billion Pakistani Rupees to provide direct cash assistance of Rupees 12000 (equal to $75 US dollar @160 = 1$) to more than 15 million families across the country. Out of 54 respondents, 27 women domestic workers or their family members received the assistance.

> My friend shared the details of the government direct cash assistance program. I sent my information. After a few hours, I received the message that I was eligible for the program. I thought it might not work but after one and a half week, I received a detailed message on mobile to visit the specific location and collect the money.

Upon further probing that majority of respondents did not get the government assistance, it was revealed that most respondents were either unaware or could not follow the process for getting registered in the system.

> When I was informed by my neighbor that such kind of cash program was launched, I tried to send my details, but it was too late.

The financial assistance support by the government was utilized by women domestic workers and their families in purchasing food items, payment of rent, and other utilities. The women domestic workers lamented over the selection process and the cash transfer mechanism. They perceived that the system in place has certain deficiencies such as available data in the system which disqualified majority of the poor families and there was no redressing mechanism for any complaint. Other than the government social support system, women domestic workers received help and assistance from charitable organizations, philanthropists, and the local community. Domestic workers received ration bags, vegetables, cooked food items, and clothes as a one-time support or help. It was also revealed during interviews that majority respondents did not ask anyone for assistance and help in the crucial time to avoid exposing their poverty and lowering their dignity; they only relied on the money they earned by working in houses, and on those house owners who provided cash and other items without even asking. Though the government assistance and social support received from different actors were considered necessary initiatives, but they were not sufficient for overall uplifting of the lives of these women domestic workers in the post lockdown situation.

## Challenges faced by domestic workers

The lockdown left poor communities, especially the domestic women workers, marginalized and vulnerable to many challenges. These challenges ranged from personal, economic, social and psychological challenges. At personal level the respondents reported stress, anxiety, and depression which adversely affected the marital relationship in many households. Conflicts at the household level were reported among couples which led to stress in marital relationship and subsequently led to domestic violence in many cases.

> My husband is a chain smoker, he needs two packets of cigarettes daily. I managed it for one and a half week but then it was difficult for me to manage such amount of money every day. My husband asked for money, but I refused many times, and told him that we are going short of money, and it was difficult to manage the household expenditures. Gradually, he became annoyed and aggressive. He attempted three times to beat me, but I luckily managed to escape.

The economic hardships resulted in the domestic workers getting laid off and suddenly losing earnings which compelled them into many compromises including cutting down on many household expenditures, food choices, health services, and pushed families into indebtedness. Due to the financial constraints, the women domestic workers restricted their social relationships and avoided participation in family and social gatherings which they perceived as adverse effects on their social standing and arrangements.

> Participation in family and social gatherings needs financial resources. But when your life is at risk and you are struggling to feed yourself and other family members, how can one participate in any kind of event? Though it negatively affects your family relationship with your relatives and others, but you have to set your priorities because once you are in a crisis, very few people will get you out of trouble.

The respondents illustrated that they were suffering from psychological problems. For instance, stress, depression, and anxiety were common issues.

> I did not pay attention to my household chores. Through I tried it several times, but I always found myself absentminded and sometimes very worried about how things will

settle down. Even my children complained so many times, but it was difficult to manage myself.

The nature of psychological problems these women domestic workers experienced vary by family arrangements. However, these psychological issues also created personal health issues among women domestic workers which they perceived were caused by the strain in the earnings and household expenditures.

## Conclusion

COVID-19 and the introduction of lockdown initiative has made all segments of society encounter an unprecedented situation which has never been experienced before. The impacts are ubiquitous and have indiscriminately affected all the sectors throughout the globe. However, those who were fragile, vulnerable and marginalized are at risk of facing adverse effects of the pandemic. The closure of economic activities in many parts of the world including Pakistan made the lives of many people difficult. Particularly, the women who were engaged in the informal sector as domestic workers to earn for their families were at higher risk to experience the pandemic at their worst. The current research endeavor documented the impact of COVID-19 on women domestic workers in Pakistan and attempted to shed light upon the dismal state of women domestic workers and their families.

Due to the sudden imposition of lockdown, the women domestic workers lamented that a chaos was created without any alternative arrangements. The closure of basic services made the life of many women domestic workers troublesome, and a severe disruption was observed in access to basic services of life. The pandemic and subsequent lockdown adversely affected the livelihood options and kept many families hand to mouth. During lockdown, these families remained impoverished and were compelled to compromise. Though they borrowed money from their relatives, neighbors and friends, yet they remained deficient in managing their household expenditures. The economic hardships such as lost jobs and sudden reduction in earnings resulted into debt which made the women domestic workers vulnerable to economic hardships. The participants struggled to readjust their household expenditures such as cutting down on food choices, access to health services, mobile and cable television recharging, rent payment and repayment of debt. Majority of the participants had lack of knowledge and awareness of the government's direct cash assistance program, therefore, minimum number of participants availed financial support. Participants considered both the government's direct cash assistance program and social support mechanism as essential initiatives but not sufficient for overall uplifting of the lives of marginalized and underprivileged segments of society. The major challenges experienced by women domestic workers were the personal issues such as strain in marital relationships, stress, and anxiety, while families restricted their social interactions and avoided participation in family and other social events which led to social isolation.

The role of women domestic workers in informal sector is crucial for the urban economy to flourish and sustain employment opportunities. The government needs to pay special attention through policy interventions that ensure job security, job structure, and inclusion of women domestic workers in social safety net programs. A contingency program that only focuses on the needs of domestic workers should be planned that can provide immediate assistance to this sector in times of crisis. In post lockdown situation, efforts should be made to restore the economic activities to the fullest potential. Particularly those business activities should be encouraged which are women inclusive and provide confidence, job security, and restoration of the earning possibilities of domestic women workers.

## Supporting information

**S1 Appendix. Verbal informed consent form for participant and demographic details of respondents in three Districts.**
(DOCX)

## Acknowledgments

We acknowledge the participation of all the women domestic workers in the data collection process. We are grateful to all the heads of households who allowed the domestic women workers to participate in a very unprecedented situation emerged from the pandemic.

## Author Contributions

**Conceptualization:** Adnan Ahmad Dogar, Ikram Shah.

**Data curation:** Adnan Ahmad Dogar, Noor Elahi, Urooj Gul Jadoon.

**Formal analysis:** Adnan Ahmad Dogar, Ikram Shah, Tahir Mahmood, Arif Alam, Urooj Gul Jadoon.

**Investigation:** Tahir Mahmood, Urooj Gul Jadoon.

**Methodology:** Ikram Shah, Tahir Mahmood, Noor Elahi, Arif Alam.

**Resources:** Noor Elahi.

**Writing – original draft:** Adnan Ahmad Dogar, Ikram Shah.

**Writing – review & editing:** Adnan Ahmad Dogar, Ikram Shah.

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
