## [Decision Letter · Decision Letter 0]

22 Aug 2022

PONE-D-22-17262Impact of Covid-19 on Informal Employment: A case study of women domestic workers in Khyber Pakhtunkhwa, PakistanPLOS ONE

Dear Dr. Shah,

Thank you for submitting your manuscript to PLOS ONE. After careful consideration, we feel that it has merit but does not fully meet PLOS ONE’s publication criteria as it currently stands. Therefore, we invite you to submit a revised version of the manuscript that addresses the points raised during the review process.

We look forward to receiving your revised manuscript.

Kind regards,

Abdulaziz Alouffi, PhD

Academic Editor

PLOS ONE

Journal Requirements:

5. We note that Figure 1 in your submission contain map images which may be copyrighted. All PLOS content is published under the Creative Commons Attribution License (CC BY 4.0), which means that the manuscript, images, and Supporting Information files will be freely available online, and any third party is permitted to access, download, copy, distribute, and use these materials in any way, even commercially, with proper attribution. For these reasons, we cannot publish previously copyrighted maps or satellite images created using proprietary data, such as Google software (Google Maps, Street View, and Earth). For more information, see our copyright guidelines: http://journals.plos.org/plosone/s/licenses-and-copyright.

6. We note you have included a table to which you do not refer in the text of your manuscript. Please ensure that you refer to Table 3 in your text; if accepted, production will need this reference to link the reader to the Table.

Additional Editor Comments:

1. Please abide by the journal guideline.

2. Check the grammar and English language editing.

3.What are statistical methodologies?

Reviewers' comments:

Reviewer's Responses to Questions

**Comments to the Author**

1. Is the manuscript technically sound, and do the data support the conclusions?

Reviewer #1: Yes

Reviewer #2: Yes

2. Has the statistical analysis been performed appropriately and rigorously? 

Reviewer #1: N/A

Reviewer #2: Yes

3. Have the authors made all data underlying the findings in their manuscript fully available?

Reviewer #1: No

Reviewer #2: Yes

4. Is the manuscript presented in an intelligible fashion and written in standard English?

Reviewer #1: Yes

Reviewer #2: No

5. Review Comments to the Author

Reviewer #1: Regarding 3, not applicable, the data is qualitative interview with human beings and therefore should not be made fully available to anyone to preserve anonymity.

The study is original, timely and relevant, and presents important findings on an under-studies sector. The qualitative approach is also relevant and adds to the understanding of perceptions and needs of domestic workers in Pakistan. Finally, this gives a strong basis for policy recommendations.

Reviewer #2: • The level of this manuscript is lower than PLOS ONE stander. However, my recommendation to the authors to increase the number of participants in future studies.

• What is the statistical software used for this manuscript? (Not clear)

• Authors should carefully follow instructions for manuscript preparation and ensure that the manuscript is proofread before submission (follow Submission Guidelines for PLOS ONE).

• This manuscript needs English editing (you should should include certificate).

6. PLOS authors have the option to publish the peer review history of their article (what does this mean?). If published, this will include your full peer review and any attached files.

Reviewer #1: No

Reviewer #2: No

---

## [Author Response · Author response to Decision Letter 0]

20 Oct 2022

Academic Editor Comments:

A separate file has been uploaded and labeled as “response to reviewers”.

2. A marked-up copy of your manuscript that highlights changes made to the original version. You should upload this as a separate file labeled 'Revised Manuscript with Track Changes'

A marked-up copy of manuscript has been uploaded that highlights changes made to the original version of manuscript.

An unmarked version of revised manuscript has been uploaded without tracked changes.

Journal Requirements:

1. Please ensure that your manuscript meets PLOS ONE's style requirements, including those for file naming. The PLOS ONE style templates.

The revised version of manuscript has followed the PLOS ONE’s style requirements.

In methodology section on line 175-179 detail has been added to address this comment by explaining the way a verbal consent has been taken from the study participants. In this study, minors were not included in data collection process. 

3. In your Data Availability statement, you have not specified where the minimal data set underlying the results described in your manuscript can be found. PLOS defines a study's minimal data set as the underlying data used to reach the conclusions drawn in the manuscript and any additional data required to replicate the reported study findings in their entirety. All PLOS journals require that the minimal data set be made fully available.

The Data Availability statement has been updated as per the PLOS ONE requirements.

The full ethics statement has been added in the Material and Methods section on line 163-167.

5. We note that Figure 1 in your submission contain map images which may be copyrighted. All PLOS content is published under the Creative Commons Attribution License (CC BY 4.0), which means that the manuscript, images, and Supporting Information files will be freely available online, and any third party is permitted to access, download, copy, distribute, and use these materials in any way, even commercially, with proper attribution.

Figure 1 has been removed from manuscript.

6. We note you have included a table to which you do not refer in the text of your manuscript. Please ensure that you refer to Table 3 in your text; if accepted, production will need this reference to link the reader to the Table.

The Table 3 has been referenced in the text on line 206.

7. Please include captions for your Supporting Information files at the end of your manuscript, and update any in-text citations to match accordingly. Please see our Supporting Information guidelines for more information.

Caption has been added to Supporting Information file as “S1 Appendix”.

Additional Comments:

1. Please abide by the journal guideline.

Manuscript has been revised as per the journal guidelines.

2. What are statistical methodologies?

In methodology section on line 188-189 the details of statistical analysis have been given.

3. Check the grammar and English language editing.

The revised version of manuscript has been proofread and edited by English language expert.

Reviewer # 1:

1. Regarding 3, not applicable, the data is qualitative interview with human beings and therefore should not be made fully available to anyone to preserve anonymity.

The study is original, timely and relevant, and presents important findings on an under-studies sector. The qualitative approach is also relevant and adds to the understanding of perceptions and needs of domestic workers in Pakistan. Finally, this gives a strong basis for policy recommendations.

We appreciate the time and efforts of esteemed reviewer for the valuable insight and feedback.

Reviewer # 2:

1. The level of this manuscript is lower than PLOS ONE stander. However, my recommendation to the authors to increase the number of participants in future studies.

In future studies the number of respondents will be increased as statistically appropriate.

2. What is the statistical software used for this manuscript? (Not clear).

The details of statistical software have been added on line 189-190.

3. Authors should carefully follow instructions for manuscript preparation and ensure that the manuscript is proofread before submission (follow Submission Guidelines for PLOS ONE).

During the revised manuscript preparation, due consideration has been paid to ensure PLOS ONE submission guidelines and proofread aspect.

4. This manuscript needs English editing (you should include certificate).

The revised manuscript has been proofread and edited by English language expert. All the changes have been indicated in “marked-up copy of manuscript” file.

---

## [Decision Letter · Decision Letter 1]

22 Nov 2022

Impact of Covid-19 on Informal Employment: A case study of women domestic workers in Khyber Pakhtunkhwa, Pakistan

PONE-D-22-17262R1

Dear Dr. Shah,

We’re pleased to inform you that your manuscript has been judged scientifically suitable for publication and will be formally accepted for publication once it meets all outstanding technical requirements.

Kind regards,

Abdulaziz Alouffi, PhD

Academic Editor

PLOS ONE

Additional Editor Comments (optional):

Reviewers' comments:

Reviewer's Responses to Questions

**Comments to the Author**

1. If the authors have adequately addressed your comments raised in a previous round of review and you feel that this manuscript is now acceptable for publication, you may indicate that here to bypass the “Comments to the Author” section, enter your conflict of interest statement in the “Confidential to Editor” section, and submit your "Accept" recommendation.

Reviewer #2: All comments have been addressed

Reviewer #3: (No Response)

2. Is the manuscript technically sound, and do the data support the conclusions?

Reviewer #2: Yes

Reviewer #3: Yes

3. Has the statistical analysis been performed appropriately and rigorously? 

Reviewer #2: Yes

Reviewer #3: N/A

4. Have the authors made all data underlying the findings in their manuscript fully available?

Reviewer #2: Yes

Reviewer #3: Yes

5. Is the manuscript presented in an intelligible fashion and written in standard English?

Reviewer #2: Yes

Reviewer #3: Yes

6. Review Comments to the Author

Reviewer #2: Regarding the manuscript ( Impact of Covid-19 on Informal Employment: A case study of women domestic workers in Khyber Pakhtunkhwa, Pakistan) the authors have adequately addressed all necessary comments , technical and questions.

Reviewer #3: The assignment haw written in an accurate way and covered all main requirements for a published paper. The main points of the paper ware mentioned in the introduction, as well as the literatures have been reviewed well, the results and discussion written in a formal way which related to the topic and linked with the literatures. The study area was motioned in material and methods section, the methodology and data collection tools have not been covered in this section properly.

7. PLOS authors have the option to publish the peer review history of their article (what does this mean?). If published, this will include your full peer review and any attached files.

Reviewer #2: No

Reviewer #3: No

---

## [Editor Report · Acceptance letter]

25 Nov 2022

PONE-D-22-17262R1 

Impact of Covid-19 on Informal Employment: A case study of women domestic workers in Khyber Pakhtunkhwa, Pakistan 

Dear Dr. Shah:

I'm pleased to inform you that your manuscript has been deemed suitable for publication in PLOS ONE. Congratulations! Your manuscript is now with our production department. 

Kind regards, 

on behalf of

Dr. Abdulaziz Alouffi 

Academic Editor

PLOS ONE